# “*More Attention than Usual*”: A Thematic Analysis of Dog Ownership Experiences in the UK during the First COVID-19 Lockdown

**DOI:** 10.3390/ani11010240

**Published:** 2021-01-19

**Authors:** Katrina E. Holland, Sara C. Owczarczak-Garstecka, Katharine L. Anderson, Rachel A. Casey, Robert M. Christley, Lauren Harris, Kirsten M. McMillan, Rebecca Mead, Jane K. Murray, Lauren Samet, Melissa M. Upjohn

**Affiliations:** Dogs Trust, London EC1V 7RQ, UK; katharine.anderson@dogstrust.org.uk (K.L.A.); rachel.casey@dogstrust.org.uk (R.A.C.); robert.christley@dogstrust.org.uk (R.M.C.); Lauren.Harris@dogstrust.org.uk (L.H.); kirsten.mcmillan@dogstrust.org.uk (K.M.M.); rebecca.mead@dogstrust.org.uk (R.M.); jane.murray@dogstrust.org.uk (J.K.M.); lauren.samet@dogstrust.org.uk (L.S.); melissa.upjohn@dogstrust.org.uk (M.M.U.)

**Keywords:** animal welfare, companion animal, COVID-19, dogs, dog walking, pet owner, SARS-CoV-2, separation-related behaviour, qualitative research

## Abstract

**Simple Summary:**

On 23 March 2020, the UK Government introduced a nationwide lockdown as part of efforts to reduce the impact of COVID-19. Lockdown control measures led many dog owners to make changes to their daily routines. This study aimed to explore how the experience of dog ownership in the United Kingdom (UK) was impacted during this lockdown. Data for this research came from open-ended survey questions on the impact of COVID-19 on dog ownership, and electronic diary entries. Three key themes emerged from the data: spending time at home with dog(s), walking practices, and behaviour and training. Owners enjoyed spending more time than usual with their dog(s) but understood that spending increased time with them may lead to the dog(s) struggling in the future when left alone. However, very few owners gave their dog(s) “alone time” during the lockdown. Owners valued the opportunity to walk their dog(s) as part of their permitted daily exercise, but some felt that lockdown restrictions on exercise did not allow them to provide their dog(s) with sufficient exercise. They also worried that lockdown guidelines restricted opportunities for dogs to interact with other dogs. Additionally, some owners noticed new problematic behaviours in their dog(s). Using these findings, we make suggestions on how dog welfare organisations might help to best support dog welfare. These include supporting dog owners in teaching their dogs to cope with being alone, even if owners do not need to leave their dogs alone.

**Abstract:**

On 23 March 2020, the UK Government imposed a nationwide lockdown as part of efforts to mitigate the impact of COVID-19. This study aimed to explore how the experience of dog ownership in the UK was impacted during this lockdown. Data for this research came from open-ended survey questions and an electronic diary completed by members of the general public and participants involved in “Generation Pup”, an ongoing longitudinal cohort study of dogs. A total of 10,510 free-text entries were analysed. Three major themes emerged: spending time at home with dog(s), walking practices, and behaviour and training. Owners valued having more time than usual with their dog(s) but also recognised that spending extra time with their dog(s) may negatively impact on the dog’s future ability to cope when left alone. However, very few owners provided alone time for their dog(s) during the lockdown. The opportunity to walk their dog(s) as part of their permitted daily exercise was regarded positively, but walks under the lockdown guidelines were not always felt to be adequate with respect to providing sufficient exercise and opportunities for interaction with other dogs. Owners reported observing new undesirable behaviours in their dog(s) during the lockdown, including barking and dogs being “clingy” or vocalising when briefly left alone. Based on these findings, we suggest intervention strategies to best support dog welfare that include helping dog owners to teach dogs to cope with being alone, even if owners do not need to leave their dogs alone.

## 1. Introduction

The infectious disease, COVID-19, caused by the newly identified severe acute respiratory syndrome coronavirus-2 (SARS-CoV-2), emerged in December 2019, in Wuhan, China. On 11th March 2020, the World Health Organisation declared a pandemic [1]. In order to control the pandemic, many countries imposed restrictions on the movement of their citizens. In the UK, following a period of advisory isolation and social distancing, lockdown measures were formally announced by the government on 23rd March 2020 [2]. Different approaches towards easing restrictions were taken across England, Scotland, Northern Ireland and Wales. In Scotland and Wales, the first wave of easing restrictions began on 11th May, when the prior ban on exercising more than once a day was lifted. This ban was lifted in England on 13th May. The first relaxation of restrictions in Northern Ireland did not occur until 18th May. Figure 1 lists key first lockdown dates across the four UK nations.

During the strict lockdown period (23rd March–11th/13th/18th May 2020 in Scotland and Wales, England, and Northern Ireland, respectively), people were only permitted to leave their home to shop for essential supplies (i.e., food and medicine), essential work (where remote work was not possible), medical needs, to provide care for vulnerable people and to take just one exercise outing away from home per day. Dog walking was not explicitly addressed in the government guidelines for daily exercise restrictions for England [15]; however, local public health officials in some areas of England did confirm that the daily exercise restrictions applied to dog walking [16]. In England, dog owners experiencing symptoms or diagnosed with COVID-19 were asked to exercise their dog(s) at home, and if not possible, arrange for someone else to walk their dog [17]. Similar guidance was implemented in Wales [18]. Guidelines for Scotland explicitly recognised that dog(s) may need to be taken out more than once a day in order to toilet, and explicitly advised shielding or vulnerable people to avoid walking their dog(s) [19]. By contrast, the guidelines for Northern Ireland asked owners to include dog walking within their once-a-day permissible exercise [20]. Across the UK, only essential travel was permitted (potentially impacting upon dog owners’ access to outdoor spaces) and owners were asked to consider keeping their dog(s) on-lead during walks to ensure they remained two metres apart from others [17].

With people instructed to “stay at home” [2], the lockdown guidelines meant that many people suddenly adopted changes in lifestyle, such as spending a significantly greater amount of time at home than they had typically done previously. We were interested in understanding the effects of these lifestyle adjustments on human–dog relationships, the perceived behaviour of dogs and practices such as dog walking. Since the first national lockdown, further COVID-19 restrictions have been variably imposed within each of the four UK nations. These are summarised in Appendix A. Additionally, many people expect to continue to work from home more permanently in the future [21]. It is therefore essential that we assess the impact these lockdown measures have had on dog owners, in order to implement appropriate intervention strategies to ensure the future health and welfare of companion dogs. Finally, this analysis will help to identify potential long-term impacts of the lockdown measures on dog behaviour. This is particularly important, as changes in dog behaviour may be long-lasting and contribute to diminished welfare as well as a breakdown of the human–dog bond, leading to relinquishment of dogs to shelters and euthanasia.

Whilst dogs were historically kept by people for principally economic or practical purposes, dogs today are more commonly understood as “pets”. Dogs provide companionship and are increasingly considered a central part of the family [22]. Indeed, many dog owners cite companionship as a key reason for having a dog [23,24]. One theory proposed to account for the popularity of pets is the idea that they serve as sources of nonhuman social support. This “social buffering” hypothesis suggests that pets act as a shield against the negative health consequences of psychosocial stress [22]. While there is considerable inconsistency regarding the mental health and wellbeing effects of dog ownership [25,26], some research suggests that pets can offer social and emotional support for adults and children and help them to cope with challenging situations [27,28,29]. The concerns and constraints brought about by COVID-19 and associated restrictions, provide an opportunity to investigate the role of pet dogs as companions, and as sources of social support within the context of a pandemic.

The authors are not aware of any research that has been conducted into the impact of routine on dog welfare. However, a review of experimental research in multiple species concluded that unpredictable environments are associated with physiological markers of stress, which, if experienced long-term, can have a negative impact on animal welfare [30]. Loss of predictability may also be more harmful to animal welfare than lack of predictability in the first place [31]. At the same time, unpredictability of events that animals find enjoyable (e.g., an unexpected additional treat or play sessions), may have a positive impact on the welfare of some individuals [30], while lack of stimulation (i.e., boredom) can be extremely aversive and impact cognitive function and welfare [32]. Consequently, organisations such as the Department for Environment, Food and Rural Affairs, Royal Society for the Prevention of Cruelty to Animals, and Dogs Trust offer guidance on how to provide sufficient stimulation (e.g., through regular walks, play, opportunities to exercise and interactions with other dogs) [33,34,35]. The guidelines also encourage establishing a routine manifested in regular feeding times, welfare-friendly handling and training, regular grooming, chances to rest and consistent sleeping habits.

Although we are unaware of research where the impact of changes in routine and dog welfare were explicitly tested, dogs’ routines and changes in their routines, in particular those leading to a dog suddenly spending more or less time with the owner, are important due to their association with separation-related behaviour (SRB) [36], which, in turn, can impact dog welfare. SRB is characterised by destructive behaviour, vocalisations and/or elimination occurring when dogs are left without human company [37,38]: signs related to anxiety are most often seen around the time of owner’s departure [37,38]. Other distress-related behaviours may also be evident, but may not be recognised as such by owners, e.g., panting, pacing, spinning, grooming, or inactivity [36]. SRB is associated with negative cognitive bias in dogs [39] and repeated anxiety during separation contributes to chronic distress and poor welfare [40]. SRB is one of the most common behaviour problems within the UK dog population [37,41,42], with an estimated 50% of dogs displaying SRBs at some point of their life [42]. This is likely to be an under-estimation of prevalence, as, by its very nature, SRB occurs when owners are out and is therefore often unrecognised. SRB is also one of the most prevalent behavioural issues recorded as a reason for relinquishment of dogs to shelters [43]. Although reasons for SRB development are not yet fully understood, it has been suggested that the lack of early positive experiences of being left alone as a puppy, and sudden changes in the routine of an adult dog that may include spending significantly more time with the owner (e.g., after being rehomed from a shelter; due to owners’ illness; holidays) and then being left alone again might be contributing factors [36]. As the lockdown restrictions impacted owners’ day-to-day activities, their capability, opportunity and motivation to care for their dogs and meet their welfare needs may have been affected. Owners spending more time at home, and additional changes to a dog’s routine, may have precipitated SRBs, both now and in the future.

This article is a part of a larger study designed to explore the impact of COVID-19 on the management of pet dogs and experiences of dog owners in the UK using quantitative and qualitative data analysis. A quantitative analysis of survey data has been published separately and highlighted a substantial increase in the percentage of owners reportedly not leaving their dogs home alone during the lockdown (14.5% prior to the lockdown vs. 57.8% during the lockdown) coupled with a decrease in the proportion of dogs left for 3 h or more at a time (48.4% prior to the lockdown vs. 5.4% during the lockdown) [44]. Other notable findings from the quantitative analysis include a decrease in the number of dogs that owners’ dogs had “met” on a typical day (8.6% of dogs met no other dogs on an average day prior to lockdown, vs. 26.3% of dogs during the lockdown). There was also evidence of an increased frequency with which people played with or did some training with their dogs during the lockdown, with the greatest increase observed in the “more than once per day” category (38.3% responded prior to the lockdown vs. 48.5% during the lockdown). Dogs were also walked less frequently (with the majority being walked once a day) compared to walking practices before the lockdown [44]. The aim of the present study was to explore how dog ownership experiences were affected by the COVID-19 lockdown restrictions in the UK, as reported by owners using complementary qualitative methods.

## 2. Materials and Methods

### 2.1. Data Collection

The data were collected via multiple methods, discussed below.

#### 2.1.1. Survey

A self-administered online survey hosted on SmartSurvey^TM^ was used to collect a convenience sample of participants. The survey gathered data on dog and owner demographics, and questioned how the lockdown restrictions had affected the dog’s routine, behaviour, health and the owner-dog bond. The quantitative data from this survey are summarised in a report published by Dogs Trust entitled “The impact of COVID-19 lockdown restrictions on dogs & dog owners in the UK” (Dogs Trust’s COVID-19 Report) [45]. Responses from two open-ended questions with free-text response fields were used in the analysis reported here. Participants were asked: (1) “what, if anything, are you particularly enjoying about dog ownership at this current time?” and (2) “what, if anything, are you particularly concerned about—related to your dog(s) or other people’s dog(s) at this current time?” Free text response fields were chosen in order to allow owners the freedom to provide in-depth answers regarding dog ownership during the lockdown. The survey was available for completion for eight weeks (4th May–3rd July 2020). This paper reports on survey data collected between survey launch (4th May 2020) and 12th May 2020, because this period covered the UK’s first strict nationwide lockdown. The survey was promoted via Dogs Trust’s social media channels (including Facebook and Twitter), a Dogs Trust e-Newsletter to supporters, an article in New Scientist magazine, and emails to dog owners participating in the Generation Pup study (www.generationpup.ac.uk) or the Dogs Trust Post Adoption study. Participants in a previous survey administered by the Dogs Trust research team who had consented to be contacted about further research opportunities were also invited to participate. The minimum age to complete the survey was 18 years and participants were required to live in the UK. An informed consent statement was provided on the first page of the study that outlined the study objectives, information that participation in the study was completely voluntary, and instructions on how to withdraw from the study.

The survey, designed as a part of a larger study, included questions which were aimed to be analysed qualitatively and quantitatively. Complementary statistical analysis of the survey data was conducted, and details of the methodological approach used and the results are presented elsewhere [44].

#### 2.1.2. Electronic Diary Responses: General Population

Electronic diary entries were collected using an online survey platform (SmartSurvey^TM^). Participants were asked about their dog’s age and whether their dog was living in their home before the lockdown restrictions came into effect. An open-ended question asked participants to describe if, and how, their dog’s life in general, and their relationship with their dog had been impacted by the lockdown. Several prompts were provided to help participants think about how the lockdown had affected them and their dog. These were related to: exercise; amount of time the dog is left alone in the home; changes in dog behaviour; veterinary care; impact of the dog on an owner’s stress levels; and the presence of any key/medical workers in the household working longer hours than usual (see Appendix B for full list of prompts included). Owners of multiple dogs could report about all dogs owned and owners could complete multiple repeat entries. The diary entry form launched on 4th May and is ongoing at the time of writing. This paper reports on general population diary data collected between survey launch (4th May 2020) and 12th May 2020. The electronic diary study was promoted in the same ways as the survey. Participants were able to complete both the survey and electronic diary aspect of the study, therefore it is possible that there was some overlap in our sample across our data sources. The minimum age to complete a diary entry was 18 years and participants were required to live in the UK and to either currently own a dog, or to have owned a dog when the COVID-19 lockdown restrictions were introduced on 23rd March 2020. An informed consent statement was provided on the first page of the study that outlined the study objectives, information that participating in the study was completely voluntary, and instructions on how to withdraw from the study.

#### 2.1.3. Electronic Diary Responses: Generation Pup Cohort

Owners of dogs enrolled in the Generation Pup study were invited to complete a diary regarding their COVID-19 experiences. Generation Pup is an ongoing, longitudinal lifetime study of dogs (including pure-breed and mixed-breed dogs), established in May 2016 (www.generationpup.ac.uk/) [46]. The online profile of each participant includes a diary where owners can note down any new, worrying or unusual experiences. Diary entries completed between 23rd March 2020 and 12th May 2020 were included in the analysis presented here. The data collection started before the general population survey, as the diary feature was already available within the Generation Pup study and starting data collection on the same day as the UK’s strict national lockdown enabled analysis of data that span across the entire lockdown period. Respondents could select the subject of the diary entry, the options included: “A new experience”, “A trip to the vet”, “An accident or worrying event”, “An illness or an injury” or “Another event or experience”. On 2nd April 2020, owners were contacted via email and reminded about the diary feature and invited to report any “COVID-19-related” experiences within their existing diary. Prompts used to help owners with completion were almost identical to those used for the general population diary study (see Appendix B for full list of prompts included). All categories of answers were included in the analysis, as even when not explicitly marked as pertaining to COVID-19, the experiences were likely to have been shaped by the lockdown-related restrictions. Owners could complete multiple entries, and if multiple dogs were enrolled in the study, they could provide entries for multiple dogs.

### 2.2. Data Analysis

Raw data were cleaned and IP deduplicated using data manipulation tools from the statistical software R (version 4.0.2; R Core Team 2016, Vienna, Austria) [47]. IP addresses were only used to subset the UK residents, and not used in further analysis. Responses to the open-ended questions and electronic diary entries were imported into NVivo (v.2, QSR) and analysed using thematic analysis. Thematic analysis is a qualitative research method for identifying, analysing, organising, describing and reporting themes or categories found within a dataset [48]. To this end, during the first coding cycle, responses were assigned coding categories which were derived inductively from the data, i.e., the assigned codes were designed to condense and summarise the raw text [49]. General population electronic diary responses were coded by K.E.H, and Generation Pup electronic diary responses were coded by S.C.O.-G. These two authors then discussed their interpretations and any discrepancies. Responses to the open-ended survey questions were coded by one of six of the authors (K.L.A., K.E.H., L.H., R.M., S.C.O.-G., L.S.), who then discussed their interpretations and any discrepancies in coding to ensure rigour and validity and to reduce the number of codes used. The coding framework was revised accordingly, and during the second coding cycle the entire dataset was re-coded using the new coding framework. These codes were then compared and grouped together to identify the main themes, i.e., over-arching insights that capture the underlying concepts [50]. Throughout the results section, direct quotes are included to give voice to the respondents and support understanding of points of interpretation. Square brackets indicate replacement of identifiable information. The main codes that were applied to the survey data were also summarised quantitatively by providing a count of times the code was applied to survey data and the proportion (as a percentage) of all main codes this represents. This was only done for the survey responses and not for the general population or Generation Pup diary entries, as the survey responses answered a more succinct question than the more general diary entries. The survey data were also based on a more representative sample of participants and responses were generally shorter than diary entries. Consequently, the number of codes is more likely to correspond to the number of survey participants who mentioned the same issue, making it easier to interpret the frequencies. However, as a single code may have been applied more than once to a response, particularly where the responses were long, the count of codes does not necessarily represent the exact number of participants who expressed a given view. Multiple sub-codes of a main code may have also been applied to an individual response. In addition, as the purpose of the analysis was to identify the diversity of dog owners‘ experiences and views, the frequency with which codes were used does not signify their importance and the numbers are not included in the main text.

### 2.3. Ethics

Ethical approval for this study was granted by Dogs Trust Ethical Review Board (Reference Number: ERB036). Generation Pup has ethical approval from the University of Bristol Animal Welfare Ethical Research Board (UIN/18/052), the Social Science Ethical Review Board at the Royal Veterinary College—URN SR2017-1116, and Dogs Trust Ethical Review Board—ERB009.

## 3. Results

### 3.1. Survey Completion

The survey results reported here correspond to the first UK lockdown, using data collected between 4th May 2020 (survey launch) and 12th May 2020 (the day before restrictions began to ease in England). The survey was started 6505 times between 4th and 12th May 2020. Following data cleaning and de-duplication, 5063 responses were deemed complete enough for inclusion within analyses. Of these, 5041 included a response to the question that asked about anything they were particularly enjoying about dog ownership at the current time, while 4922 contained a response to the question that asked about anything they were particularly concerned about, related to their dog(s) or other people’s dog(s). Of those who provided a response for their gender (*n* = 4306, 85%), 86% (*n* = 3684) were female and 14% (*n* = 616) were male. Less than 1% (*n* = 27) preferred not to say or provided an “other” response. Age categories of respondents who responded to this question (*n* = 4314, 85%) are summarised in Table 1.

### 3.2. General Population Electronic Diary Completion

Sixty-six diary responses were collected from 64 participants between 4th and 12th May 2020. Sixty-two respondents completed one entry each, whilst two respondents completed two entries each. Respondents were mostly female (*n* = 57, 89%). Six were male and one did not disclose this information. Age categories of respondents are summarised in Table 1.

### 3.3. Generation Pup Electronic Diary Completion

Four hundred and eighty-one diary entries were collected from 262 participants (i.e., 109 responses were completed by repeat diarists, some of whom completed more than two entries) between 23rd March and 12th May 2020. Of these, 271 diary entries from 195 participants were explicitly described as related to COVID-19 experiences. Thirty-seven participants provided more than one COVID-19-related diary entry within the period of analysis. Respondents were mostly female (*n* = 240, 92.3%). Eighteen participants were male and two did not disclose this information. Age categories of respondents are summarised in Table 1.

### 3.4. Qualitative Themes

This study explored how human–dog relationships were affected by the first COVID-19 lockdown restrictions in the UK. In particular, we investigated how the lockdown affected owners’ experiences of dog ownership and owners’ perceptions of their dogs’ experiences during this period. Reporting on survey and electronic diary data together, these experiences are described here by discussing three key themes that emerged: spending time at home with dog(s), walking practices, and behaviour and training (summarised in Table 2).

#### 3.4.1. Spending Time at Home with Dogs

Many owners reported spending more time at home during the lockdown and described enjoying more quality time with their dog, valuing dog companionship as a crucial form of social and emotional support during the lockdown. Amongst our respondents, dogs reportedly provided their owners with love and emotional support, helping them to cope with the concerning situation:
“*Life at home would be very lonely without my dogs, and [dog names] are a great comfort as they are such loving dogs. It never feels as if I am living a solitary life.*”(Generation Pup respondent, 5th April 2020).
“*The dogs have had a huge influence on this time. Given a focus, a reason to get [up] in the morning and have made the whole situation bearable.*”(General population diary respondent, 5th May 2020).

Some respondents explicitly linked their dog ownership with benefitting their mental health during this period:
“*The mental health benefit of owning a dog has [n]ever been so apparent.*”(Survey respondent, 7th May 2020).
“*She is helping me with anxiety and depression, which is getting worse in this pandemic.*”(Survey respondent, 12th May 2020).

A valuable aspect of dog ownership that contributed to perceptions of improved mental wellbeing was the distraction that the dog provided:
“*Distraction of caring for a dog rather than getting stressed/worried about the situation.*”(Survey respondent, 7th May 2020).

Importance was placed on tactile interactions, such as cuddling, between owner and dog. Some owners directly associated the significance of their dog’s companionship with the temporary inability to have tactile interactions (e.g., hugs) with family or friends:
“*I think living alone [dog name] has helped to keep me sane. I’m really missing contact with family & friends. But it would’ve been so much worse without having [dog name] here. He is a snuggly dog so he is great company.*”(General population diary respondent, 12th May 2020).

The company of dog(s) was particularly important to owners who lived alone, with respondents suggesting that their dog(s) mitigated against loneliness:
“*I live alone and for me having my dog has made a huge difference. Just having another living being to cuddle and hang out with, massively alleviated a sense of isolation.*”(Survey respondent, 5th May 2020).

Owners also reported feeling less guilt or worry as they no longer had to leave their dog home alone whilst they went out to work, which may contribute to some owners’ reluctance to enforce alone time:
“*[U]sually I feel guilty leaving them to go to work.*”(Survey respondent, 5th May 2020).

As many people had more time to spend at home, respondents expressed concerns that others may consider the lockdown to be a good time to acquire a dog, without considering the long-term commitment of dog ownership and how they would be able to care for a dog when they returned to work. Some also worried that dogs acquired during the lockdown might develop future behavioural problems, for example, due to difficulty socialising puppies at this time, and dogs becoming used to constant company that might not be sustained as the lockdown eases. Consequently, there were fears about potential relinquishments following the lockdown:
“*I know a few people who have got puppies meaning well and thinking it is a good time but I think the lack of socialization and being used to people being around all the time might not be the best thing for the dogs longer term and I hope those people have thought it through.*”(Survey respondent, 5th May 2020).

The lockdown did not affect all dog owners’ routines in the same way. Whilst many were spending extra, or all their time, at home, a small number of owners explained that their dog’s routine had not changed:
“*Nothing has changed. We are still able to go for walks on the beach. Usually see the same very few people/dogs.*”(Survey respondent, 5th May 2020).
“*I don’t work so I’m with my boy 24/7 so this is no different than any other day for him.*”(Survey respondent, 6th May 2020).

A few people mentioned that their dog(s) were currently spending more time alone than in the past. For example, in the case of this key worker:
“*I’m working on the frontline so am going to work as normal but am no longer able to come home at lunch so [dog name] has had to get used to being left all day (about 8 h).*”(Generation Pup respondent, 11th April 2020).

While many respondents reported that the lockdown had eased their worries about having to leave their dog home alone, the lockdown brought about other fears, especially in relation to access to veterinary care:
“*I am very worried about her health as she was due to have an operation on her heart on [date]. It was cancelled the day before as it was not considered urgent, at that point. I am worried that she may suffer further heart damage because of the delay, which is not helping my stress levels.*”(Survey respondent, 11th May 2020).
“*Seriously anxious about anything happening to my 15 year old [dog]. I have always assumed I would be with her to support her at the end, but I fear that may not now be allowed. I just pray she stays ok until vet practices are back to normal.*”(Survey respondent, 5th May 2020).

In addition, a handful of owners worried what would happen to their dog in the case of the owner’s illness, hospitalisation or death, and discussed their contingency plans (or lack thereof):
“*For my dog I worry in case I catch the disease and had to go to hospital. Her welfare is so important to me therefore I worry that she would be looked after properly if I’m not there.*”(Survey respondent, 9th May 2020).

#### 3.4.2. Walking Practices

Respondents generally reported that the opportunity to walk their dog during the lockdown was highly prized and had a positive psychological impact on their wellbeing. Some cited their daily walk as the best part of their day, during the strict lockdown phase, when people were permitted to take only one exercise outing per day:
“*Taking my dog out for a walk is the highlight of my day at the moment.*”(Survey respondent, 7th May 2020).

For many owners, the need to walk their dog provided them with a daily routine and contributed to their sense of purpose during this uncertain time:
“*I wouldn’t leave the house at all if I didn’t have to take my dog for a walk.*”(Survey respondent, 7th May 2020).

In addition, the opportunity for more “family walks” was regarded positively by many owners. In many cases, the lockdown measures meant that members of the family who were typically unable to join in everyday walks (e.g., due to work commitments away from home) were now able to. The daily dog walk was also considered a motivator to encourage children outside:
“*Dog walks is one genuine reason to leave the house which the whole family can contribute to.*”(Survey respondent, 5th May 2020).
“*My husband who normally misses out on all weekday walks is really enjoying it.*”(Survey respondent, 5th May 2020).

Some owners mentioned a further benefit of walking: the opportunity to see and engage with other people whilst on the walk, highlighting another way in which dog companionship, and human–dog activities such as dog walking, may have helped to provide emotional and social support during the lockdown:
“*Local dog park gives an opportunity for socially distant interaction with other owners, helps to feel part of a community.*”(Survey respondent, 7th May 2020).
“*The only people I get to speak to are on our daily walk.*”(Survey respondent, 5th May 2020).

The physical environment in which owners walked was mentioned, with some owners reporting that they were enjoying walking in quieter places. In addition to encountering fewer people or dogs, some noted that less traffic on the roads contributed to a more peaceful walking experience compared to typical walking experiences:
“*Our walks seem quieter. Less traffic, less people. It’s been nice.*”(Survey respondent, 7th May 2020).

Respondents frequently commented on changes regarding the amount of exercise their dog(s) undertook during the lockdown but varied in their accounts of these reported changes. Some owners reported having more time to walk, and therefore walking the dog for longer than usual. In some cases, this meant that walks during the lockdown were more leisurely than usual and felt less of a chore:
“*Able to enjoy longer walks in the morning without the pressure of having to go to work/start for the day.*”(Survey respondent, 7th May 2020).
“*Walks are a pleasure rather than before when sometimes on busy days it was a chore.*”(Survey respondent, 6th May 2020).

However, others felt the official restrictions impeded their ability to exercise their dog appropriately. Many respondents reported that dogs were exercised less often and their walks during the lockdown were not as frequent as usual:
“*Also feel bad that walks have reduced from 3 per day to 1 due to restrictions.*”(General population diary respondent, 5th May 2020).

A small number of owners stopped taking their dog(s) for a walk altogether, due to a fear of infection or illness:
“*I’m also afraid that they would pick any virus in their coats and bring it home, so we aren’t going for walks any longer, just exercising at home.*”(Generation Pup respondent, 6th April 2020).
“*I am a nurse and have tested positive to COVID virus. I have been very ill for 11 days but luckily able to stay at home as I am improving. [Dog’s name] has noticed I’m ill and even though he loves his walks I haven’t been able to go.*”(Generation Pup respondent, 4th April 2020).
“*Not feeling comfortable about taking him out for walks due to the Covid 19 situation.*”(General population diary respondent, 12th May 2020).

Some respondents also had concerns about their dog’s health deteriorating as a result of changes in their dog’s exercise routine. For instance, one owner observed their dog gaining weight:
“*[My] main concerns are that [dog’s name] isn’t getting enough exercise and socialisation; that [dog’s name] will get overweight (He is already slightly heavier).*”(General population diary respondent, 12th May 2020).

A small number of owners of arthritic dogs mentioned that restrictions on walking were having a negative impact on their dog’s health:
“*My dog is elderly, has arthritis. She needs little and often walks but I am only supposed to go out once a day with her (till 13th May). She normally has weekly hydrotherapy but can’t do this at the moment. Her legs have noticeably deteriorated during the lockdown.*”(Survey respondent, 12th May 2020).

Many owners who would have typically driven to preferred walking locations stopped doing so, opting for more local walking routes instead. Some enjoyed this aspect of their change in routine, as it enabled them to explore new local areas:
“*Look forward to ur [sic] walks together and taking exercise as we explore new places locally to walk—I love to see him really enjoy his walkies.*”(Survey respondent, 12th May 2020).

Many owners negotiated the social distancing guidance (to stay at least two metres away from others) and the needs of other public space users (e.g., runners, cyclists and other dog walkers) by keeping their dog(s) on a lead more than usual. Additionally, in some spaces, new rules required that dogs be kept on-lead. However, some owners were concerned that this did not provide a good enough walking experience for their dog:
“*She can be frustrated on lead and normally likes to play around with other dogs off lead.*”(Survey respondent, 5th May 2020).
“*Seldom able to let them off lead for a real gallop.*”(Survey respondent, 5th May 2020).

Furthermore, walks during which the dog spent much of the time on-lead were not, by some, considered “proper” walks:
“*I keep him on a lead more and haven’t had the chance to go on a ‘proper walkie’ for weeks.*”(Survey respondent, 5th May 2020).

Concerns were also raised about the perceived potentially detrimental impact of on-lead walking for dog’s social behaviour and training:
“*We are now required to keep her on a lead in all our usual parks, which may set her back in terms of recall.*”(Survey respondent, 5th May 2020).
“*Our concern is when he is eventually able to be petted by a stranger or run off lead again or play with other dogs.*”(Survey respondent, 7th May 2020).

To overcome the travel restrictions and their dogs’ need for exercise, some owners walked dog(s) in different locations to usual. However, seeking quieter local walking routes or changes in the time of day dog(s) were walked (compared to usual) may have led to some locations being busier, as many respondents reported. Crowded spaces were linked to experiences in which attempts to socially distance were challenging or stressful. Some owners were dissatisfied with other walkers, who they perceived as seeking more desirable walking locations and coming into “their” areas. Some also thought that others’ reasons to be outside walking dog(s) were not genuinely linked with promoting the dog’s welfare, and thus judged their reason to be outside as less valid:
“*It’s quite stressful even walking her while trying to avoid other people as my husband is vulnerable as he has CKD [chronic kidney disease].*”(Generation Pup respondent, 6th April 2020).
“*Our local park has been overrun by people who, frankly, have overweight and hellishly unsocial/untrained dogs who I doubt have been exercised more than twice a year. It seems like dog ownership has become the “key to going out” but that people haven’t ever really invested what is needed to have a dog that can be out in public.*”(Survey respondent, 11th May 2020).

Other dog walkers’ behaviour was further criticised by respondents, with many dog owners noticing more dog faeces left uncollected:
“*I am seeing a lot more dog mess than I have before. It appears as though more owners are taking advantage of walking their dogs (which is good!) but not cleaning up the waste (which is bad).*”(Survey respondent, 5th May 2020).

For owners of dogs with prior behavioural issues, such as anxiety or reactivity around other dogs, walking experiences during the lockdown were variable. For some, walks during the lockdown were more enjoyable, with dogs and owners reportedly benefitting from quieter spaces and social distancing:
“*The quieter roads/pavements & the fact people are moving away & keeping their distance when we walk—my younger collie can be reactive—mostly to other dogs, so it’s nice to have more space than normal!*”(Survey respondent, 7th May 2020).

However, some owners reported increased difficulty walking their anxious dog(s), linking this to busier walking routes and an increase in unfamiliar dogs they were encountering:
“*One of my dogs is anxious around other dogs and can be reactive on the lead. I have found that there are lots of different dogs about at the moment that I would not normally come across during our daily walks and this has made walks difficult and added to his anxiety.*”(Survey respondent, 7th May 2020).
“*All the extra people out around 6pm who are suddenly compelled to take daily exercise are stressing out my middle dog, who is a nervous rescue. She stopped wanting to go for an evening walk for a few weeks, she only wanted to go out in the morning with my parents.*”(Survey respondent, 5th May 2020).

As a result, for some, dog walks had become less recreational in nature and more functional instead. For instance, some long or leisurely walks were replaced with local “street-walks” or short “loo walks”. Some owners worried and questioned whether these walks were sufficient in providing adequate exercise for their dog(s), hinting that variety is needed:
“*I do feel guilty going on the same walks every day!*”(Generation Pup respondent, 3rd April 2020).

Respondents were concerned about how their dog(s) felt about the changes to their walks, fearing their dog(s) were “frustrated” or “bored”. Walks had also become stressful and a cause for concern to some owners due to other walkers allowing their dog(s) off-lead and not keeping their distance:
“*To be honest, taking her for a walk is stressful because of the difficulty of managing social distancing alongside trying to keep her focused on me when we reach the Common (to continue short exposures to it as part of training her to focus on us rather than the environment). It also means that I don’t get a decent walk myself unless we stick to roads and pavements where she is fine and settled.*”(Generation Pup respondent, 3rd April 2020).

However, some respondents were sad that their dog was no longer allowed to play with other dogs during walks:
“*Concerned about the lack of social interaction he’s receiving, especially with other dogs. He is very friendly, and is struggling not being able to go and say hello/play.*”(Survey respondent, 11th May 2020).

While some owners’ walking practices had been significantly affected by the lockdown, a number of owners reported no changes to dog walking or other daily routines due to the pandemic:
“*We are very fortunate in that COVID has had no real impact on our daily routine and our lives. My husband and I are both retired so both at home all the time. We live on a [very large] private estate with just [a small number of] homes, ample places to walk and meet friends with dogs which I still do twice daily.*”(Generation Pup respondent, 15th April 2020).

As the respondent in the above excerpt indicates, access to suitable exercise areas and convenient walking locations is likely one factor that might affect owners’ experience of exercising and walking a dog under the lockdown conditions. The impact on daily routines more generally was also variably affected by employment status (e.g., working from home, working outside the home, retired, furloughed), as the above excerpt also highlights.

For those who felt that their dog’s exercise or enrichment opportunities outside the home had been impeded by the lockdown restrictions, walks were sometimes supplemented with play in the garden or the use of enrichment games, as they felt it necessary to compensate for changes in walking routine:
“*We’ve also been keeping them entertained with ‘find it’ games using the recycling boxes and tubes etc. This can tire [dog’s name] out totally.*”(Generation Pup respondent, 11th April 2020).
“*We live on a shared site (…) By profession I am a dog walker so we are normally out walking all day. We have probably only done about 7 walks off site in all that time [since lockdown restrictions were imposed on 23rd March 2020], the rest just playing frisbee games in the garden.*”(General population diary respondent, 11th May 2020).

#### 3.4.3. Behaviour and Training

##### Behaviour Linked to Changes in Daily Care Routine

Both positive and negative changes in dog behaviour were widely reported. In some cases, behavioural changes were associated with adjustments to a dog’s daily routine. Dogs who had previously been left home alone for at least some of the day were, in some cases, reported to have constant company. These changes to the dog’s routine were largely interpreted by the owners as positive from the dog’s perspective. Respondents typically reported that they believed dogs were enjoying the increased company:
“*My dogs are loving us all being at home.*”(General population diary respondent, 6th May 2020).
“*Lots [of dogs] have more attention than usual which surely can’t be a bad thing.*”(Survey respondent, 8th May 2020).

At the same time, some owners reported their dog(s) showing behavioural changes towards owners, such as clinginess:
“*She turned in to a Velcro dog at first, following me to the point where I was tripping over her.*”(General population diary respondent, 8th May 2020).

There were also reports of some dogs barking or jumping when their owner left, or prepared to leave, the house or even a room:
“*[Dog name] is being very dramatic when we have to leave the house without her, barking, leaping up at us and the door.*”(General population diary respondent, 5th May 2020).

In addition, many more were worried about their dog, or dogs in general, developing issues around separation post-lockdown, when more “normal” routines resume, for example, leaving the dog whilst owners attend work. Owners of dogs who had shown signs of separation-related behaviour (SRB) prior to the lockdown were particularly concerned:
“*I also worry that his separation anxiety will get worse and dread the day that he is left alone again in the home.*”(Survey respondent, 8th May 2020).

Owners were also anticipating their dog feeling lonely and taking time to adapt to post-lockdown routines in the future:
“*[I am] worried he is so used to having us around 24/7 that it will take time for him to adjust when we have to go back to work and leave him alone for small periods of time.*”(Survey respondent, 11th May 2020).

Although many owners were aware that their dog may struggle adjusting to being alone again, or with at least reduced human company, only a small minority of owners reported that they were implementing training around separation:
“*[W]e are consciously making an effort now to leave him alone so he is not used to company all of the time.*”(General population diary respondent, 11th May 2020).

Rather than maintenance of “alone time” being the norm amongst owners, many instead anticipated needing to “re-train” their dog in the future:
“*I am here all day everyday now. We noticed even if we have left him in a room alone while we do something (…) he becomes distressed more quickly than he ever would do before. He will also follow us round the house at all times. I worry for what long-term effects this will have on his happiness—suppose we’ll just have to re-train all the good habits we worked so hard on in the first place!*”(Generation Pup respondent, 9th April 2020).

The increased time many owners had been spending with their dogs helps to explain why they were not enforcing “alone-time”:
“*Now [dog name] has not been left on his own since lockdown. We have become too anxious to consider leaving him at home, even for short periods, because it has been so long. We know that we need to re-start crate training and reducing his separation anxiety ready for when we return to working in an office.*”(General population diary respondent, 6th May 2020).
Owners also reported a shift in who was caring for the dog. As well as adults working from home, respondents mentioned children being home-schooled, who were thus spending more time with the dog and, in some cases, helping out with walking. Some owners felt their dog’s behaviour had improved as a result of this increased company from multiple caretakers:
“*[Dog name] we think has benefitted from us being around in terms of behaviour.*”(General population diary respondent, 8th May 2020).

Many who had recently acquired puppies or rescue dogs considered the “extra time” afforded by the lockdown to be beneficial for settling their dog in:
“*Our most recent dog, from [name of rescue organisation], is a very nervous dog with a difficult past and having extra human company at home is really helping her begin to gain confidence and start to learn to play. It’s very rewarding.*”(Survey respondent, 8th May 2020).

While many dogs were described as enjoying this extra attention, some described their dog(s) as being unsettled and restless:
“*Since mid-March I have been working from home, and she spends all day sleeping by my make-shift desk. When this first started she would sleep soundly until about 4pm, when she’d start to stir, being properly awake at 4.30 when I would normally arrive home. Now if I’m late having my lunch I get a [dog breed] nose reminding me it’s time for some [dog name]-time.*”(General population diary respondent, 5th May 2020).

Attention-seeking behaviours were not reported as frequently as reactivity, clinginess or reaction to being separated from owners, but were common nevertheless:
“*[Dog name] would be alone in the house for up to 9 h quite happily whilst I was working. Since lockdown I’m here all the time. He doesn’t understand that when I’m at the dining table on my laptop that I can’t just play with him or snuggle on the sofa. This has led to him sometimes whining or barking at me when I’m on conference calls. He will sometimes paw at my arm to get my hand away from the laptop. So I have had to instigate some training around this area.*”(General population diary respondent, 12th May 2020).

Some dogs were also described as being more destructive:
“*[Dog name] who is fast approaching her first birthday, on [birth date] has gone from a quiet well behaved [dog breed], into a ‘Deva’ [sic], being destructive and demanding attention.*”(General population diary respondent, 5th May 2020).

Though rare across the dataset, some owners reported instances of their dog(s) mouthing, nipping or biting:
“*[Dog name] used to play gently before lockdown—he would get excited but never bite or nip. Since we’ve been home all the time he bites and mouths so much, worse than when he was a little puppy and was testing out his teeth (…) [Dog name] has also developed a new behaviour since lockdown, where he growls and grumbles at my partner when he walks past [dog name] bed. I could understand this if [dog name] had been woken from sleep abruptly, but he is always awake and alert when this happens (…) He is not a grumbly or growly dog usually.*”(General population diary respondent, 6th May 2020).
“*I have [additional adults] and a [teenager] as well as myself and my husband in the house. The dogs are not always coping well with all the over stimulation with us all here, especially late in the evening. One has started to get a bit snappy with some members of the household and has actually bitten one of my sons. This is really out of character for her so we have been trying to give her much more quiet time with just one of us at a time in our bedroom.*”(Survey respondent, 11th May 2020).

Alongside these new undesired behaviours, owners expressed concerns about their dog’s emotional wellbeing, often referring to their dog’s perceived state of boredom or depression:
“*I have noticed that [dog’s name] is a bit down and not as active and motivared [sic] as usual.*”(Generation Pup respondent 6th April 2020).
“*He still seems bored and frustrated by lack of exercise.*”(Generation Pup respondent 4th April 2020).

##### Behaviour Linked to Reduced Social Interaction

According to our respondents, attempts to socially distance alongside keeping dogs on-lead had led to a reduction in dogs’ opportunities for social interactions with both dogs and people on walks and in other contexts. Owners believed that through social contacts, dogs acquire (in the case of puppies) or maintain their social skills with other dogs and with people. Some owners also thought that dogs need to experience other objects or experiences (e.g., bikes, cars, traffic) to get used to them. Additionally, many owners believed that their dog was missing social contact with other familiar dogs or humans, which they were worried may have a negative impact on their dog’s emotional wellbeing, by making dogs frustrated or bored:
“*He is a people dog and loves to interact with them, but he is not able to get close to people and that frustrates him.*”(Survey respondent, 5th May 2020).

Many respondents were worried about how the lack of social contact with other dogs or people may impact on dog behaviour and training plans:
“*[Dog’s name] was only able to go on one walk after having all her vaccines before lockdown hit. Her socialisation has suffered as a result. (…) I hope we do not end up with behavioural problems as a result of the lockdown.*”(Generation Pup respondent, 16th April 2020).
“*He isn’t seeing visitors of course and as we are still working on his excitement around visitors that will be a challenge.*”(General population diary respondent, 5th May 2020).
“*One of our dogs is scared of people, not being able to socialize her is not good for her progression.*”(Survey respondent, 7th May 2020).

Many owners linked the development of new undesirable behaviours when seeing other people or dogs, such as barking at passers-by and other dogs or pulling on the lead, with reduced social interaction with people and dogs:
“*[H]e has become stressed and reactive around other dogs and people. He does not lunge or bark (he never did) but does pull towards people/dogs with great enthusiasm. We think he is missing interactions with people/dogs.*”(General population diary respondent, 6th May 2020).

These concerns were particularly prominent among those living in more remote areas, where contact with other people and dogs was minimal and further reduced by the lockdown restrictions, who reported that their dog(s) were being more reactive towards non-social phenomena, such as sound or traffic:
“*We live in a small, quiet village which has now become even quieter. She has often reacted to noises by barking but has definitely become far more reactive since lockdown. She will bark (severely bark) at any noise—and not stop easily.*”(General population diary respondent, 5th May 2020).

Missing out on social interaction was particularly stressful for those in single-dog households and owners of young puppies. The latter were concerned about the loss of opportunities during a critical period in their puppy’s development to socialise with other dogs and people, and their puppy acquiring “life skills” through being exposed to novel sounds and objects:
“*My dog is a young puppy, he should be starting to learn socialising skills, meeting new people and dogs too.*”(Survey respondent, 10th May 2020).

Some puppy owners adapted socialisation techniques to make the most of their current situation:
“*We will be using time in the car to let her see people. Come and go and treats to try and de sensitise her to people.*”(Generation Pup respondent 16th April 2020).
“*We are still walking our other dog each morning for about an hour and I carry [dog’s name] round so she is used to the sounds and experience.*”(Generation Pup respondent 3rd April).

However, not all owners were aware how they may continue to give their dogs a range of experiences under the lockdown conditions, e.g., by exposing them to noises, traffic and different people (including those with hats, beards, umbrellas, etc.).

##### Impact on Training

As well as affecting dog behaviour, respondents also reported that the lockdown had impacted training. Some owners of dogs with pre-existing behavioural problems (e.g., those described as anxious, fearful or under-socialised around people or other dogs before the lockdown) expressed that a structured training plan they were working on prior to the lockdown, which aimed at addressing a dog’s problem behaviour (a ‘behaviour modification plan’), had been interrupted:
“*He is showing anxiety (barking & lunging) towards approaching people. Not other dogs, thank goodness. This started just before lockdown and is not resolving. (…) I so wish I could let him go and say hello to a friendly people and before lockdown he did bark a bit but if allowed to approach he was absolutely fine and friendly. I know what I should be doing but am so frustrated I am not allowed to!*”(General population diary respondent, 12th May 2020).

Our findings indicate that many owners have the knowledge, but not the opportunity or motivation, to address and prevent their dog’s behavioural issues under the lockdown conditions. Many owners are aware of the importance of early life socialisation, and the ongoing need to maintain interactions with familiar people and dogs, but were not able to achieve this during lockdown. To compensate for this, they focused more on dog training (obedience/tricks and games).

While the lockdown restrictions hindered organised group training classes, a number of owners mentioned that their training classes were still being carried out online. Nonetheless, due to social distancing, some felt it was harder to practice dog “life skills”. However, whilst several owners were worried that their dogs’ “life skills” may be deteriorating, in some cases obedience and trick training had improved, as respondents reported having time to do more training at home, sometimes incorporating games or using online resources:
“*I’ve had so much more time to focus on his general obedience and training. He’s learned some new tricks and is observing things like boundaries and door manners much more since I’m working on them everyday.*”(General population diary respondent, 6th May 2020).
“*She is doing lots of scent work and basic training and trick training. We have been working on games too. [Dog’s name] is learning to play treat noughts and crosses.*”(Generation Pup respondent, 8th April 2020).

As well as seeing improvements in their dog’s learning, many owners reported that training their dog was an enjoyable human–dog activity undertaken during the lockdown:
“*We are doing some training with the younger dog and it’s really good fun!*”(Survey respondent, 6th May 2020).

## 4. Discussion

It is clear from our study that the COVID-19 lockdown control measures affected dog ownership experiences in a number of diverse ways. In particular, restrictions on leaving the home (for work and exercise) and social distancing guidance impacted household routines, dog walking practices and experiences, and opportunities for social interaction with both other dogs and humans. These changes were consequently perceived to impact dog behaviour. Here, we return to our findings and discuss them in the context of previous research.

### 4.1. Reported Changes in Dog Behaviour

Lockdown restrictions that led to owners spending more time with their dogs may have had a positive impact on the welfare of many animals. Whilst the length of time individual dogs can be left alone varies, depending on factors such as age and training, experts recommend that dogs are not left alone for more than four hours [51]. Despite this professional guidance, recent research suggests that, of an estimated 9.9 million pet dogs living in the UK, 19% are left alone for five or more hours every day [24].

Some owners noticed their dog(s) being more “clingy”, vocal or distressed when left alone, and more eager to seek proximity to, and attention from, their owner. The changes in lifestyle linked with COVID-19 may influence the subsequent risk of SRBs when dogs are left alone again [36]. Moreover, the potential development of SRB in the future was a significant cause for concern amongst respondents. The qualitative findings presented here are in line with findings from the quantitative data collected as a part of the same survey [45]. Dogs Trust’s COVID-19 Report identified a 41% increase in dogs being “clingy” towards adults and 46% increase in dogs being clingy towards children compared to before the lockdown [45]. The quantitative analysis of additional data from the survey reported on here, published elsewhere, indicated a four-fold increase in the number of dogs not left at home alone for longer than 5 min (an increase from 14.5% prior to the lockdown to 57.8% during the lockdown) [44]. Previous research conducted amongst pet owners during the COVID-19-related lockdown in Spain also reported results that suggest an increase in dog behaviours that might be associated with SRB [52]. Amongst Spanish dog owners, 41.6% reported their dogs were displaying more attention-seeking behaviour during the lockdown. Therefore, it appears that changes in dog behaviour around owner attention and leaving the house are common between countries facing lifestyle changes due to altered working practices, such as an increase in working from home, redundancies and furlough. In our study, some owners expressed awareness that spending more time with their dog may have a negative impact on the dog’s future welfare, either due to the development of SRBs or, more generally, the dog struggling to adapt to the post-lockdown routine. Therefore, these dog owners had the capability required to prevent SRBs, as they had knowledge of how changes in the routine may impact the dog. However, most owners, in general, did not enforce their dog’s normal routine or “alone time” and thought of it instead as something they should address in the future.

The “COM-B” model of human behaviour may be helpful in explaining discrepancies in owner awareness of changes in their dog’s behaviour but failure to take remedial action. In this model, a behaviour is an outcome of a person’s *capability* (i.e., having the necessary knowledge and physical skills and other characteristics), *opportunity* (i.e., the environmental or systemic attributes that together with person’s capability make the behaviour possible) and *motivation* (i.e., all mental processes that ignite the behaviour) to perform it [53]. Following from the COM-B model, it is plausible that as owners enjoyed their dog’s companionship and often relied on the dog’s presence for social and emotional support, they were not motivated to enforce “alone time”. Feeling guilty about leaving the dog alone and fearing how a dog may behave when left alone, may have also impacted upon owners’ motivation. In addition, the size and layout of the home, restrictions on outdoor activity for non-essential purposes, as well as the number of household members isolating, could have made the enforcement of “alone time” difficult, thus impacting on the owner’s opportunity to make the necessary changes to their behaviour.

As SRB may impact the human–dog bond, leading some owners to relinquish their pets or seek euthanasia [54,55], it is crucial that owners: (1) understand the importance and feasibility of preventing SRB during the lockdown, (2) know how to teach their dog “home alone” skills, and (3) have the opportunity and motivation to do so. As post-lockdown routines are likely to make home-working more common, communication with owners should be sensitive to the emotional support and enjoyment that owners derive from spending more time with their dog and address their motivation for preventing SRBs.

In addition to SRBs, owners believed that changes in dogs’ routines led to dogs being more restless, unsettled, having disrupted sleep, being more vocal, attention-seeking and destructive. Owners linked these behaviours with reduced variety of activities, insufficient exercise and stimulation. Furthermore, they were worried about dogs being, or becoming, bored. These changes are similar to those described in Dogs Trust’s COVID-19 Report which identified an 82% increase in dogs whining or barking when someone was working or busy, in addition to an increase in other attention-seeking behaviours [45]. Findings described in the same report also identified an increase in behaviours associated with fear and frustration in dogs, such as hiding, grabbing items of clothes, snapping and nipping during play and when approached or touched. This is similar to the research findings from Spain where owners also reported dogs showing more common behavioural problems and undesirable behaviours during the lockdown, with excessive vocalisations and fear of loud noises being particularly common [52]. The same study also reported that 24.9% of respondents felt that their dog(s) had become more nervous during the Spanish lockdown, 20.8% more excitable, 18.4% more frustrated, 16.4% more stressed, and 11.3% more relaxed. Although research into animal boredom is scant, boredom could be expressed in a multitude of different ways. These may include reduced activity, increased drowsiness during the day, sleep disruptions and restlessness, as well as signs of frustration or seeking sensation (e.g., by seeking attention, showing increased exploratory or destructive behaviours) [32]. Lack of stimulation could, therefore, explain some of the observed behaviours in both this study, Dogs Trust’s COVID-19 Report [45], and the research reported from Spain [52]. However, with most respondents spending increased time with their dog(s), it is possible that the perceived increase in undesirable behaviours was linked to the owners being with their dog(s) more, and were hence more likely to observe the behaviour.

Whilst some owners reported providing dogs with more stimulation in the form of training or games, many owners potentially lacked capability and/or opportunity to do so. As boredom is suggested to be aversive [56], it is important to improve owners’ knowledge and skills. Resources, such as those produced by Dogs Trust [35,57,58], suggest how owners could provide enriching activities for dogs during future lockdowns or periods in which they may be largely confined to the home for other reasons.

### 4.2. Walking Practices and Experiences

Many respondents in our study enjoyed walking their dog(s) during the lockdown, citing the positive impact that this activity and the routine it provided, was felt to have on their mental wellbeing. This finding supports previous research that found dog walking was positively associated with owners’ mental wellbeing, through its enjoyable and stress-relieving qualities [59]. Fewer time constraints and access to and quality of the physical environment enabled longer and more leisurely walks for some respondents. In addition, with many adults working from home and most children being home-schooled, the lockdown also facilitated walks that involved more members of the household, referred to as “family walks”. In these ways, the lockdown generated what Westgarth et al. (2020) have termed “recreational” dog walking for some owners, in place of “functional” walks [60].

Spending time together—on walks and otherwise—is important in establishing a stable bond with a dog and has been assessed in questionnaires measuring the strength of the human–dog bond [61]. Walking with a dog during the lockdown created the opportunity for shared emotional experiences and reciprocal interactions for owners, possibly contributing to the development or maintenance of affectional bonds.

Nevertheless, concerns surrounding dog walking also emerged in our findings. This is in line with previous findings that Spanish dog owners’ most common concern about the effects of the lockdown was prohibition of dog walking [52]. Similar to the experiences of owners affected by the national lockdown in Spain [52], for many of our respondents, the experience of dog walking had been affected in several ways which meant that walks were less pleasant under lockdown conditions and had become more “functional”. One influential factor was the reduced, or, in some cases, absent, opportunity for interaction with other dogs or people. Diminished choice of walking locations, due to travel restrictions and local park closures, crowded public spaces replete with other dog walkers, and the owner’s dog(s) being kept on-lead also contributed to this type of walking experience. Previous evidence indicates that crowded spaces can act as a barrier to walking for some dog owners, as they feel it impedes their walking with their dog off-lead [62,63,64]. In addition, factors linked with the behaviour of an individual’s own dog(s) may also contribute to the experience and likelihood of a dog being walked, e.g., dogs with behavioural problems being walked less often [23,65,66]. In our study, owners of dogs with pre-existing behavioural problems that are challenging outside of the house, such as reactivity to other dogs or people, reported mixed walking experiences. This finding suggests that walking experiences were likely shaped by a combination of factors in addition to their dog’s behaviour, for instance, how busy their environment was and whether other people and dogs were maintaining their distance.

At the beginning of the lockdown people were initially restricted to one form of outdoor exercise per day, potentially reducing both the frequency and total duration of daily dog walks. Our results highlighted diversity in the effect of this restriction on dog walking practices. Many reported that their dog(s) received fewer daily walks during the lockdown. Physical exercise has benefits for the health and fitness of the dog, and can also provide important opportunities for environmental enrichment [67]. Reduced frequency and duration of walking dogs, as a result of exercise restrictions during the lockdown, could have the potential to reduce the amount of mental stimulation dogs receive if efforts are not made to provide additional enrichment, for example, through play and/or training. Although many reported a reduction in the frequency of daily dog walks, the total duration of daily walking was not necessarily affected. This is in line with the analysis of additional quantitative data of from our survey, reported on elsewhere [44]. Whilst there was a significant reduction in the number of daily walks for many dogs, the reduction in daily walk duration from prior to during the lockdown was less marked, though still statistically significant. Particularly where respondents were spending more time at home than usual, for instance, those who do not typically work from home, some reported having more time to dedicate to dog walking. However, as a result of the “once daily” rule, some reported replacing multiple daily walks with a single, longer walk. Meanwhile, some others reported an increase in the frequency of walks their dog(s) were given. In some multi-adult households, people who do not usually perform dog walking duties, e.g., partners who do not typically work from home, were reported as doing so during the lockdown. This meant that some dogs were provided with their usual, or even greater, frequency and/or total duration of daily walking. Our results thus suggest that household composition may be a factor to explore further in relation to walking frequency and duration during the lockdown. This finding is consistent with the quantitative analysis of our additional survey data, in which it was found that households with more than one adult were significantly less likely to report a reduction in walk frequency [44]. However, previous evidence investigating the association between having more people in the household and daily dog walking behaviour was inconclusive overall [68].

Some of our respondents voiced concerns about whether their functional walks provided adequate exercise for their dog, both in terms of the dog’s health and fitness and the amount of mental stimulation they were receiving. Research into the relationship between exercise (including walking) and dog welfare is surprisingly scant and limited to animals in rescue shelters. For dogs in shelters, additional calm-handling and exercise sessions resulted in some changes in behaviour indicative of positive dog welfare [69] and physiological changes suggestive of lowered stress [70]. However, walks can be very variable, in terms of length, frequency, time off-lead and opportunities to run or play and engage in dog or human social interaction. It is unclear what makes a walk sufficient exercise for pet dogs, and this is likely to vary between individuals. Owners’ perception of functional walks being insufficient for their dogs may therefore highlight a welfare concern and/or reflect owners’ own level of enjoyment or satisfaction with this type of walking. In addition to concerns about the amount of exercise they were able to provide, many owners reported their dissatisfaction with being unable to visit preferred walking locations, such as physical environments considered more “exciting”. However, our findings demonstrated that for some other owners, the opportunity to explore previously undiscovered walks in their local area was valued. This is in keeping with previous research that has demonstrated the importance of the physical spaces in which dog walking occurs, with owners wanting to walk in environments that induce positive emotions for both themselves and their dog(s) [60]. Within this previous study, spaces that facilitate exploring were identified as one aspect of the physical environment which can shape such positive emotions. Another important element of the physical environment that can have a positive impact on the experience of dog owners is the environment’s provision for off-lead exercise [60]. Our results found that the ability to access open spaces where owners felt confident to let their dog(s) run freely off-lead, contributed to walking experiences. Consistent with Westgarth et al. [60], on-lead walks were generally perceived as “boring” for the dog.

Studies have illustrated that dog owners understand dog walking as an expression of their care for the dog, with an individual dog’s perceived needs and preferences found to shape dog-walking decisions [71,72]. For instance, walking locations are chosen with a dog’s interests and physical capabilities in mind [71]. Consequently, within the current study, owners’ experience of on-lead walks as unsatisfying could be understood alongside such evidence that has identified care as fundamental to shaping decisions surrounding dog walking, such as where, when and how dog walking takes place [71]. Understanding our findings alongside this evidence, we suggest that owners’ worries about changes in dog walking practices may reflect a feeling of being unable to provide the same quality of care for their dog(s) during the lockdown that they would under normal conditions, with the dog’s preferences and capabilities in mind. It is known that the ability to walk a dog off-lead is important for many dog owners [73]. Restricting dogs to more, or only, on-lead walking constrains the dog’s capability to act independently and may impact their freedom to express natural dog behaviours, e.g., sniffing or investigating at their leisure. Owners reported dogs having fewer opportunities to be “dog-like” and engage in activities like sniffing or greeting and playing with other dogs. The perception that keeping a dog on-lead constrains the dog’s enjoyment of walking could contribute to owners feeling they are unable to provide care that meets the dog’s needs and preferences [72].

Our findings suggest that the lockdown led to an increased pressure on dog walkers to negotiate spaces shared by others: a feature of dog walking in the UK that previous research has highlighted [72]. As the UK does not have “dog parks”, like those common in USA or Australia, walking spaces are shared with other users, including cyclists and joggers, as well as other dog walkers. During the lockdown, respondents were typically keen to avoid interacting with others during dog walks. This behaviour is not unique to the lockdown, as previous findings have also identified how some walkers actively attempt to avoid others, for instance, choosing quiet walking routes [72]. Previous research also suggests that factors surrounding other people’s dogs, for instance the fear of encountering other aggressive dogs are negatively associated with dog walking [73]. However, it is likely that the social distancing guidelines and more crowded local walking routes contributed to a greater need to negotiate spaces, as even fewer people desired social interaction during this period. Our findings also point to a culture of judgment towards other dog walkers, particularly those perceived as “new” or “inexperienced”, with respondents especially concerned about others not keeping their dogs under control. This is in line with previous research that identified a culture of judgment amongst dog walkers in England, in which non-regular walkers were actively shunned from the local dog walking community [72].

Whilst dog walking during the lockdown prompted considerable negotiation of the spaces shared with others, in some cases, the activity did encourage social contact between people, in line with findings from previous research [74]. In these instances, dogs were acknowledged to act as social facilitators who can enable “contact, confidence, conversation and confederation among previously unacquainted persons who might otherwise not spoke” [75] (p. 23). Our findings suggest that, for some owners, this social aspect of dog walking had been impeded by social distancing measures during the lockdown. However, for others, walking the dog during this period continued to provide an opportunity to speak to other people. For the latter, encountering other people during walks fostered a sense of community between dog walkers: another function of dog walking that has been recognised in previous research [76].

As Degeling and Rock (2012) note, “‘[p]laces’ and ‘practices’ connect people and animals” [71] (p. 405) not only by creating opportunities for social interaction, but also by increasing chances of spreading zoonotic diseases. Respondents’ desires to avoid interactions with other dogs were associated in part with concerns about the potential epidemiological role of dogs in the spread of COVID-19: fears that were expressed worldwide in the early stages of the COVID-19 outbreak [77,78,79]. A study by Westgarth et al. (2010) found that the use of a lead reduced the frequency of dog–dog interactions during walking, leading the authors to conclude that on-lead walking could be a potentially effective modifier of dog interactions, helping to reduce infectious disease transmission [80]. Although the anecdotal reports of COVID-19 spread from pets to people have not been verified by experts, many of our respondents endeavoured to avoid other dogs during walks by keeping their dogs on-lead and some passed judgment on other owners who chose not to do so.

In summary, our findings regarding dog walking suggest a tension experienced by dog owners who, from an infectious disease perspective, were largely keen to avoid other dogs, especially those off-lead. However, at the same time, from a social perspective, many owners felt dissatisfied that their dogs were unable to enjoy freely interacting with other dogs. This reduced opportunity for socialisation is discussed in greater detail in the following section.

### 4.3. Social Interaction

Our findings highlighted owners’ concerns regarding the impact of social distancing measures, practice of on-lead walking, and restrictions on the number of walks, on dogs’ ability to interact with other dogs and people. Lockdown-related changes led to a reduction in dogs’ exposure to other dogs, people, and everyday phenomena and experiences, such as traffic noise, car journeys and cyclists. The quantitative analysis of our additional survey data found a significant decrease in the number of dogs that owners’ dogs had “met” (i.e., been in the same room, or within 2 m if outside) on an average day [44]. There was a three-fold increase in the number of dogs that met no other dogs on an average day (8.6% before the lockdown vs. 26.3% during the lockdown). Owners were concerned about how this would impact the social behaviour of adult dogs, which they believed was crucial for dogs to retain their “life skills”. Owners also discussed how the lack of social interactions with familiar people and their dog’s “friends” (both canine and human) was felt to be having a negative impact on their dog’s welfare, as they thought dogs were missing their human and canine friends. Owners of young dogs and puppies acquired shortly before or during the lockdown worried that their dog may miss out on primary socialisation and thus never develop crucial “life skills”. This concern is in line with previous research which highlights the importance of puppy socialisation for the prevention of future behavioural problems. Fear of other dogs and unfamiliar people has been associated with poor socialisation during puppyhood (defined as a number of encounters with other dogs and people between 7–16 weeks) and infrequent participation in training and other activities (which could be a proxy for contact with other people and dogs) [81]. In addition, puppies and young dogs that had attended classes showed less aggression towards familiar dogs, were rated as more trainable, less sensitive to touch and showed less non-social fear a year later than those who had not attended classes [82]. Attendance at puppy classes was also found to be associated with a reduced risk of aggression to unfamiliar people entering the house [83]. However, in another study, no association between attending puppy classes and later aggressive behaviour towards unfamiliar dogs was identified [84]. Beyond any potential impact on later behaviour towards other dogs or people, lack or absence of appropriate puppy socialisation to various situations and environments may contribute to fearfulness associated with novel situations, loud noises (e.g., fireworks) and different walking surfaces [85]. Worries surrounding the impact of restricted socialisation motivated a small proportion of puppy and young dog owners to break the social distancing and/or exercise rules to socialise their dogs and/or to look for alternative ways to enable social contacts.

### 4.4. Human–Dog Relationships

Our findings suggest that dog ownership potentially provided psychological and mental health benefits to owners during the lockdown, echoing previous research which suggests that dogs can improve the general wellbeing and mood of owners [86]. Using validated dog-owner relationship scales, Bowen et al. (2020) identified that changes in lifestyle linked with the COVID-19 lockdown in Spain led to a strengthening of bond with dogs [52]. In particular, emotional closeness and interactions with dogs had increased, while the perceived costs of dog ownership had decreased. Regarding the level of support obtained from their pet, the same study reported that 47% percent of respondents stated that their pet had helped them moderately more or significantly more throughout the confinement compared to before. In particular, the study identified that female pet owners and those who had a stronger bond with a dog had higher odds of experiencing greater support from their dog(s).

Our results also highlight the potential for dogs to help mitigate against feelings of loneliness. This finding is in line with previous research conducted amongst Australian pet owners living alone during the COVID-19 lockdown, whereby dog ownership was found to protect against loneliness [87]. The authors of the Australian study suggest this buffering effect may be linked to the presence of a physical connection with another being—i.e., having another living creature one can touch. Our findings support this interpretation, as for many of our respondents, a notable positive aspect of their time spent with their dogs was cuddling.

### 4.5. Strengths and Limitations

This study provides the first qualitative analysis of UK dog owners’ experiences of dog ownership during the COVID-19 lockdown. However, this work is subject to several potential limitations. Our analysis is based on owner reports and perceptions of behaviours, and therefore may not be as accurate as direct clinical observations of behaviour. The owner’s relationship with their dog(s) may have shaped their experiences and perceptions of dog behaviour. Previous research suggests that the likelihood of reporting that a dog’s behaviour is deteriorating increases for owners who rate their emotional closeness to a dog as high, owners who report themselves as often “getting mad” with a dog, and owners who report increased overall general changes in their routine [52]. Similar links were identified in a study of pet ownership during the lockdown in Israel, where dog owners’ perceptions of quality of life were linked with their judgment of their dogs’ quality of life and emergence of new behavioural problems [88]. Furthermore, lack of owners’ comments on a given behaviour is not equivalent with a dog not showing these behaviours. Previous research has highlighted that even when dogs are reported to show behaviours indicative of SRB, owners may not consider such behaviours as problematic [89].

The count of codes reported in Table 2 is also limited. Whilst the counts are included as an indicator of the prevalence of a given code, the frequency of occurrence does not necessarily indicate its significance, as the aim of this analysis was to explore the diversity of views and experiences, rather than to estimate their frequency.

In addition, the inconsistencies in the lockdown rules across the four UK nations, and in particular different guidelines on dog walking, may have impacted on walking-related experiences in ways that our analysis did not account for.

Data collection for all three methods used in our study did not begin at the same time. Our earliest data (Generation Pup diaries) were collected throughout the entire lockdown period, due to this feature being already available to the Generation Pup cohort. However, as we only reminded Generation Pup participants to complete diaries regarding COVID-19 experiences on 2nd April, most of the COVID-19 specific entries come from the period of time starting eleven days after the lockdown began. The general population diaries and survey were not available for completion until 4th May. It is, therefore, possible that our data may not have captured a complete range of lockdown experiences during the first six weeks of the lockdown. However, our study is strengthened by having some data available during the early stages of the lockdown.

Finally, as our study used a convenience sample, it may suffer from sampling bias due to self-selection by participants. Participants who took part in this study were also primarily female. However, this makes our findings comparable to previous research, as most studies of human–pet relationships are subjected to over-representation of female participants [90,91,92]. Nonetheless, as our findings are based on a large dataset (*n* = 10,510 free-text entries), the experiences and challenges discussed here are likely to capture sentiments of the UK’s dog owning population. We have also demonstrated how they echo the lockdown experiences of dog owners in another country [52].

## 5. Conclusions

The current study explored how the UK’s nationwide COVID-19 lockdown impacted dog owners and their perceptions of the impact of COVID-19 on their dogs’ behaviour and welfare. Through thematic analysis of electronic diary entries and free-text responses to survey questions, three main themes emerged: time spent at home, walking practices, and behaviour and training. These findings have implications for the design of interventions aiming to promote dog welfare, particularly in relation to preventing SRBs. While many owners were concerned about how their dog(s) would adjust to being left alone in the future, “alone time” was not typically being enforced. Generally, owners were enjoying spending more time with their dog(s) than usual and felt supported by their dog(s) during this time. Emphasis on training dogs to cope with “alone time” as an act of care towards dogs may encourage owners to either begin or continue to train their dog(s) to spend time alone. In addition, this message could be conveyed by highlighting how a dog’s ability to cope when left alone for short periods of time contributes to their life skills, welfare and a better human–dog relationship. Further work to improve owners’ knowledge of socialisation techniques and permissible practice under lockdown conditions could also be beneficial. Clear and accessible guidelines on how to ensure that puppies and adult dogs can be exposed to a range of different experiences whilst minimising risk of virus transmission could help owners make informed decisions. As some owners reported the behaviour of their dog(s) deteriorating, provision of behavioural first-aid for owners who are noticing a change in their dog(s) behaviour could also be helpful. Further research is needed to investigate short- and long-term behavioural changes in dogs during and after the lockdown.

## Figures and Tables

**Figure 1 animals-11-00240-f001:**
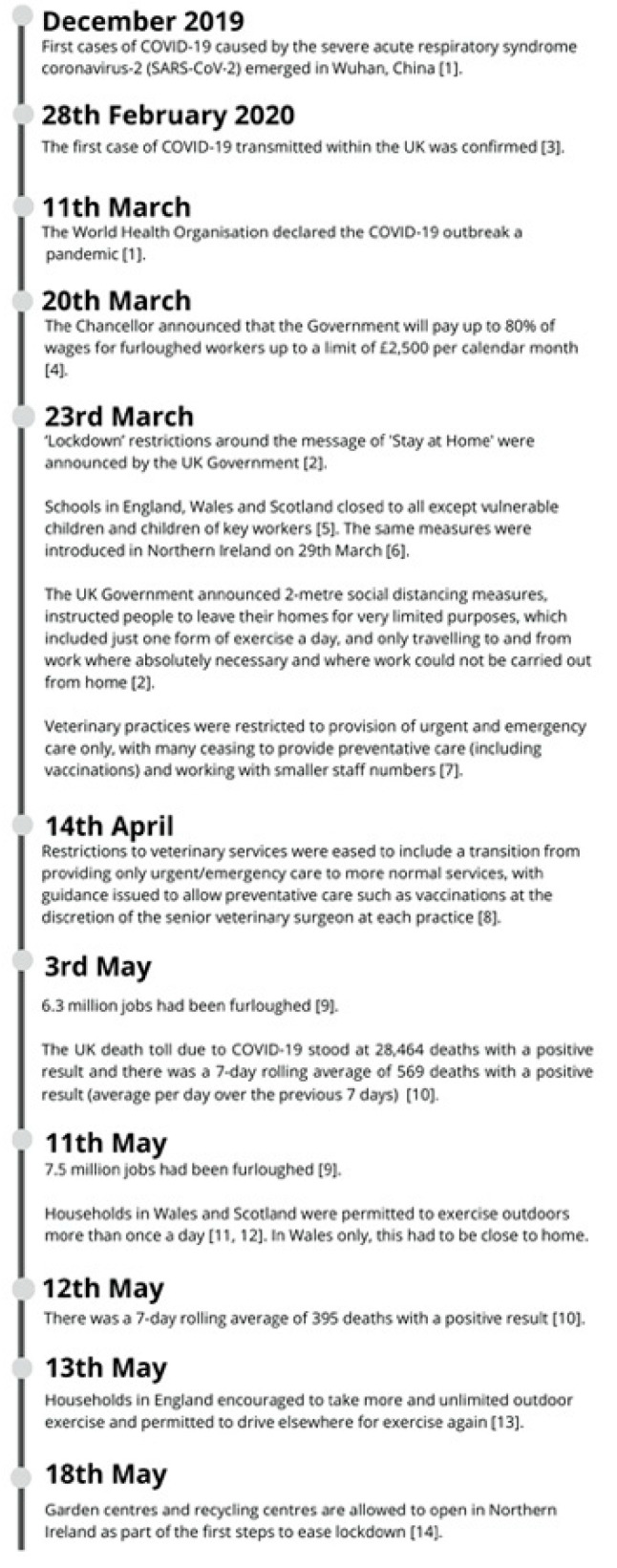
Statistics and dates relevant to this study across the four UK nations [1,2,3,4,5,6,7,8,9,10,11,12,13,14].

**Table 1 animals-11-00240-t001:** Age of Survey, General Population Electronic Diary and Generation Pup Electronic Diary Respondents (n, %).

Age Category (Years)	Survey Respondents	General Population Electronic Diary Respondents	Generation Pup Electronic Diary Respondents
18–24 ^†^	169 (3.9)	2 (3.1)	7 (2.7)
25–34	648 (15.0)	7 (10.9)	33 (12.6)
35–44	592 (13.7)	9 (14.1)	40 (15.3)
45–54	1046 (24.3)	15 (23.4)	66(25.2)
55–64	1088 (25.2)	20 (31.3)	77(29.4)
65–74 *	632 (14.7)	8 (12.5)	39 (14.8)
75–84	127 (2.9)	3 (4.7)	n/a
85 years or older	12 (0.3)	0 (0.0)	n/a

^†^ This age category was defined as 16–24 in the Generation Pup Electronic Diary. * The oldest age category in the Generation Pup Electronic Diary was age 65+.

**Table 2 animals-11-00240-t002:** An overview of major themes and codes identified in the course of analysis. The number of times each main code was applied to survey response data is provided (*n*) alongside a percentage which reflects the proportion of the main codes that each main code accounts for (%). The number and percentage calculations are based on the survey content only; the main codes and sub-codes were, however, present also across the general population and Generation Pup diaries.

Theme	Main Codes	*n*	%	Examples of Sub-Codes
Spending Time at Home with Dogs	Owner Health and Wellbeing	1621	12.8	Improves owner’s mood Mental health benefits Distraction Provides purpose Source of comfort Love the dog provides
	Companionship	1869	14.8	Mitigates against loneliness Friendship
	Concerns Over Other People’s Dogs	512	4.1	Concerns for dogs acquired during the lockdown Fear of dogs being relinquished after the lockdown
	Concerns for Dog’s Welfare and Medical Care	857	6.8	Dog spends more time alone (e.g., key workers) Reduced availability of veterinary care Fear of being unable to care for the dog due to owner’s illness
Walking Practices	Enjoyment of Dog Walking	1356	10.7	More pleasant walking experiences ○ Walking a dog as highlight of a day ○ Walking a dog provides routine ○ Family walks ○ Walks as an opportunity for social interactions (human–human) with other people ○ More leisurely walking experience ○ Exploring local area ○ Easier, where quiet, to manage dogs with existing behavioural problems
	Changes in Dog Walking and Exercise Routine	2180	17.3	Less pleasant walking experiences ○ Fear of COVID-19 transmission during walks ○ Impact of restricted walking on dog’s health/welfare ○ Socially distanced walks as not “good enough” ○ Difficulties in maintaining social distance on dog walks ○ Challenging, where busy, to manage dogs with existing behavioural problems on socially distanced walks Supplementing/replacing walks with exercises/enrichment at home
Behaviour and Training	Behaviour Linked to Changes in Daily Care Routine (-,1974, 27.21)	1725	13.7	Dogs enjoying increased human company Signs of separation-related behaviours Fear of separation-related issues developing in the future Concerns over dog’s ability to adapt to post-lockdown routine Strategies for preventing separation-related behaviours in the future Anticipating a need to re-train a dog in the future Attention-seeking behaviours Restlessness/difficulties in settling down Increase in destructive behaviour (when householders were present) Increase in mouthing/nipping Concerns over dog’s emotional wellbeing
	Behaviour Linked with Reduced Social Contact	981	7.8	Dogs missing social contact with other familiar dogs or humans Concerns over the impact of lack of social interaction on dog’s future behaviour Undesirable behaviours as a result of limited social interaction Concerns over a puppy not having a chance to develop social skills Socially distanced methods of socialising puppies
	Positive Impact on Training	379	3.0	More time for training New games and training plans Enjoyment of at home training during the lockdown
	Negative Impact on Training	204	1.6	Interruption of ongoing training Interruption of ongoing efforts to address behaviours owners find problematic
No Concerns *	n/a	940	7.4	
No Aspects Owner is Enjoying †	n/a	9	0.1	

* “No concerns” code was applied to all survey responses in which the participant reported that they had no concerns. This code was only applied to data from survey participants. Of the 5041 survey responses included in analysis, an additional 119 participants left this question response field blank, but provided a response to the question about aspects they were enjoying. † “No Aspects Owner Is Enjoying” code was applied to all survey responses in which the participant reported that they were not enjoying anything, or nothing in particular, about dog ownership during the lockdown. This code was only applied to data from survey participants. Of the 5041 survey responses included in analysis, an additional 22 participants left this question response field blank, but provided a response to the question about aspects they were concerned about.

## Data Availability

The data presented in this study are available on request from the corresponding authors. The data are not publicly available due to ethical approval of participant informed consent that included survey respondents being informed that we will remove all personally identifiable information before sharing data with Universities and/or research institutions.

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
