# Peer review of "More Attention than Usual”: A Thematic Analysis of Dog Ownership Experiences in the UK during the First COVID-19 Lockdown"

_animals, 2021, doi:10.3390/ani11010240_

Round 1

Reviewer 1 Report

This is a very interesting manuscript focusing the attention on how Dog Ownership Experiences during the first COVID-19 Lockdown in the UK.

Due to its theme, it is certainly of great interest, innovative and original. This manuscript provides important information on potential emerging problems in the pet-owner relationship and management related to the COVID-19 pandemic.

The approach used for data collection allows gaining a varied picture of the situation, even if based on the opinions and experiences of the owners. The authors adopted an innovative approach to analysing the data: the Thematic analysis. It has been quite interesting to me because honestly, I am not an expert in this kind of analysis, therefore I am not able to give any comment on how rigorously the method has been applied and on the quality of the selection process and coding reliability. So, I will not even attempt to assess the methodological approach just assuming that the authors made the best.

On other hand, it would have been interesting to have, even as supplementary data, a statistical analysis of the data evaluating the importance and ratings of certain information collected, to what extent some problems are shared among the owners and how they are related to particular types of animals, their modalities of management or to the characteristics of the owners, including the period in which the data was collected in relation to the lockdown, and so on…). This could give more emphasis to the results in strengthening their value. As a reader, I gain no knowledge of how much the different statements discussed and the problem experienced by the owners are shared and spread in the sample of the population tested. Even if this was not the objective of the manuscript, it surely would add value to the great amount of data collected.

The discussion is well and clearly articulated, on general concepts related to specific results obtained from the elaboration of the data collected by the Survey and by the two Electronic Diary Responses.

Other suggestions:

I would suggest adding information reported on Line 83-89 in the Supplementary Material (from "For Example, England…..)

Could you please better describe the method used to identify the three key themes that emerged: spending time at home with dog(s), walking practices, and behaviour and training (summarized in Table 2)?

While greatly appreciating the work done by the authors, also given the large quantity and quality of the data collected, I would strongly suggest the authors add to their work a comparative analysis of the data to strengthen the new knowledge arising from this beautiful work and fully enhance its scientific importance.

With the hope that my invitation to make the work more complete will be welcomed as an encouragement to improve it, I await with great curiosity and interest to read the revised version of this manuscript.

The terms used in the results section: “Owners, many or some owners” without any quantitative data analysis can be very roughly interpreted. Would be out of contest report in bracket descriptive data?

Congratulation for your very interesting work.

Reviewer 2 Report

The manuscript: “More attention than usual”: A Thematic Analysis of  Dog Ownership Experiences in the UK during the  first COVID-19 Lockdown, raises an important topic of the impact of the introduced COVID-19 lockdown on the welfare and behavior of dogs kept in houses in the Great Britain. This is a problem wich is seen in different countries around the world. The aim of this study was to explore how dog ownership experiences were affected by the COVID-19 lockdown restrictions in the UK, as reported by owners.

Comments:

Line 109 – “Little research has been conducted into the impact of routine on dog welfare.” – It is not true, dogs function perfectly in the so-called routine, i.e. conditions of repetition of daily activities, they feel safe and know what to expect because daily schedule is repeatable and therefore predictable. Every behaviorist, zoopsychologist and dog trainer knows this.

The transition to line 122 on Separation-related behavior (SRB) appears here quite unexpectedly, without any previous reference to this type of behavior, and seems out of context. At least one introductory sentence would be welcome here.

Various research tools were used in the methodology: Survey questionnaire. Diary Electronic Diary Responses: General Population and Electronic Diary Responses: Generation Pup Cohort. This may not be a big mistake in this case, but the authors describe that in those tools included other questions regarding the lockdown situation, and this is a methodological error, especially since the respondents could use both Survey and Dairy. The authors themselves write that the data sources are repeated (line 185 – “Participants were able to complete both the survey and electronic diary aspect of the study, therefore it is possible that there was some overlap in our sample across our data sources”.)

Line 160 – “The survey was available for completion for eight weeks (4th May - 3rd July 2020). This paper reports on survey data collected between survey launch (4th May 2020) and 12th May 2020.)”. – So why only the data from the period May 4 to May 12, 2020, i.e. 9 days, were used - it is necessary to explain why.

In the second research tool: Electronic Diary Responses: General Population, the time of data collection was the same as in the case of surveys. (line 183 This paper reports on general population diary data collected between survey launch (4th May 2020) and 12th May 2020.)

However, data from the third Electronic Diary Responses: Generation Pup Cohort tool was collected in a different period - from April 2 to May 12, 2020 (line 203 Diary entries completed between 2nd April 2020 and 12th May 2020 were included in the analysis presented here.)

It was possible to use research tools from the same period, because the time of data collection overlaps, but for an unknown reason, the time of collecting data from the latest Electronic Diary Responses: Generation Pup Cohort study was 41 days, not as in the case of Survey and Electronic Diary Responses: General Population 9 days. An explanation should be given as to why and whether  there was no alternative.

Entry options for Electronic Diary Responses: General Population and Electronic Diary Responses: Generation Pup Cohort are standardized (Appendix A), while the survey entries included two questions that respondents could answer by entering a detailed text (line 156  Participants were asked: (1) ‘what, if anything, are you particularly enjoying about dog ownership at this current time?’ and (2) ‘what, if anything, are you particularly concerned about - related to your dog(s) or other people’s dog(s) at this current time?’ Free text response fields were chosen in order to allow owners the freedom to provide in-depth answers regarding dog ownership during the lockdown.) However, it is not clear whether any other questions were included in the survey, especially compatible with those hints contained in Appendix A.

However it is the chapter “Results” and the way of compiling the results that raise most concerns. The results are described in a popular and not scientific way, there are no numerical or percentage statements or statistical analysis. And you can group the collected results according to the diagram of topics in Table 2 and elaborate them statistically, present them graphically in charts to make them legible. The whole chapter Results is very scattered, long; quoting the answers of respondents by the authors disturbs the reception of the work and becomes tiring for the reader after only a few pages of the text, and we basically do not learn what was the distribution of these answers in relation to the entire work.

For example, line 303 “The company of dog(s) was particularly important to owners who lived alone, with respondents suggesting that their dog(s) mitigated against loneliness” – it can be considered as one of the results in order to  determine how many people think so, or one can also divide it into responses obtained from individual research tools (Survey, Electronic Diary Responses: General Population and Electronic Diary Responses).

Line 318-125 – there is an important thought here, how many people considered the lockdown as a good time to acquire a dog, without considering the long-term commitment of dog ownership and how they would be able to care for the dog when they go back to work.

As well, responses are important which express concerns that these dogs may develop behavioral problems in the future due to the difficulties in socializing of the puppies in this time and the continued presence of the owner at home. What will happen to these dogs after lockdown is complete? The answers to the questions (line 318-325) tell about the knowledge of dog owners about their behavior and possible disorders, so it is even necessary to provide numerical values here.

The introduction of additional subsections (3.4.1. Spending Time at Home with Dogs, 3.4.2. Walking Practices, 3.4.3. Behavior and Training (and here are three additional sections)) to the Results chapter does group the cited answers thematically but this data should be numerical and contained in a table or graph, possibly described without these quotes. In such work, the authors have to use specific numbers or percentages and not just vocabulary: an overwhelming number of owners, or many, some, most - it doesn't say anything, such phrases can be used in Conclussion and not in Results.

Line 857-863 unnecessary introduction. It is generally known that the Disscusion Chapter is there to compare your results with the research of other authors. However, when there are no numerical values, how can such a comparison be made? And the authors compare their results (which are not really there) to the percentages obtained by other authors, so they could follow the works they cite (Dogs Trust The impact of COVID-19 lockdown restrictions on dogs & dog owners in the UK; 2020; i Bowen, J.; García, E.; Darder, P.; Argüelles, J.; Fatjó, J. The effects of the Spanish COVID-19 lockdown on 1370 people, their pets and the human-animal bond. J. Vet. Behav. 2020, doi:10.1016/j.jveb.2020.05.013.)

There are also many repetitions of Resuls in Discussion that are often not relevant to the research done by other authors, e.g. the entire paragraph 1107-1133 belongs basically to results, not the discussion.

Line 1154 “This study provides the first qualitative analysis of UK dog owners’ experiences of dog ownership during the COVID-19 lockdown.” – very exaggerated, especially since the authors, apart from quotes from the surveys, do not support their research with a statistical study or even data expressed as percentages.

The subject of the manuscript is very valuable; however, it is impossible to draw conclusions on the basis of this study, as there are no numerical values that would document the study.

As it stands, the work is more of a story than a research article and is not suitable for publication in Animals in this form. It is unfortunate, because the number of responses collected from the respondents is very big and could therefore be representative.

Reviewer 3 Report

This is an excellent paper. It is very clearly written, edited and the study described. It is a lengthy paper and could typically be shortened, but I don't think this is necessary given the need for context with emerging COVID-19 related findings. Further, I am not convinced that there is a need for two quotes each, makes for a long paper, but it is such a novel area to understand that it is good to have.

A few minor questions:
Line 85 – What is a firebreak lock-down?
Line 97 – awkwardly stated
Introduction – very well written, with all of the context necessary to place the study focus. One figure was very stark, and may want to consider adding it to the abstract – decrease in the %age of dogs left alone for 3 hours or more (from 48% to 5%).
Line 173 – Why the time period May 4 - 12 for the electronic diary entries?
Line 203 – Used a broader range of dates for the PUP cohort (April 2 to May 12).
It may be useful for the reader to share in summary why the varied dates for the survey and two electronic diaries?  If this is a limitation of the study share this upfront, and then return to it later identifying why it was a limitation and how you could or could not overcome it.

Reviewer 4 Report

It is a very interesting topic, but the design of the study is quite simple and poor from a methodological and statistical point of view. This research and its datashit must be re-analyzed with the support of statisticians, otherwise, this big amount of data has just poor scientific value.

Round 2

Reviewer 2 Report

In the redrafted Manuscript:  “More attention than usual”: A Thematic Analysis of  Dog Ownership Experiences in the UK during the  first COVID-19 Lockdown, corrections have been made which improve the quality of manuscript. However…

The authors found no studies of dogs’ functioning  in routine conditions -  Line 110 comment in Review 1, did not cite other authors, and did not change the sentence (“Little research has been conducted into the impact of routine on dog welfare.”).

The chapter Results (3.1, 3.2, 3.3) clearly shows the percentage breakdown by gender of the respondents and their total number in responses using individual research tools.

However, in subsections 3.4.1, 3.4.2 and 3.4.3, the authors chose not to introduce numerical or percentage ranking of the results obtained, leaving quotations and imprecise phrases: “many, some owners, small number of owners ”. And it was this part of the results that my remarks were addressed. There are still no numerical or percentage statements or statistical analysis in this part. Quotations from the respondents' statements have not been reduced and they in no way enrich the work in such number.

Key corrections and additions have not been introduced. The manuscript has not really been rewritten, only minor revision have been made.

The form of citing respondents' statements is not recommended in scientific journals, and in such an amount. This is especially true of journals that are not in the humanities profile. If the editors of the journal Animals accept this extensive form of citation, it is absolutely necessary to supplement the results in a percentage or numerical form. The work could then be published with major revision in the Results chapter and reduction of citations.

Reviewer 4 Report

The topic is very interesting, but the methodology is far too imprecise. The statistical analysis is really unsufficient, due to the methodology. Multivariate analysis with better structured questionaire, would have been much more precious. There is too many anecdotal information (citations from questionaires).
